# Establishment of a Rapid and Precise Nutritional Screening Method for Convalescent Rehabilitation Patients: A Preliminary Study

**DOI:** 10.3390/nu16233997

**Published:** 2024-11-22

**Authors:** Kozue Okamoto, Miho Kogirima, Yoshiro Tsuji, Shinsuke Ishino, Hiromasa Inoue

**Affiliations:** 1Department of Food Science and Nutrition, Doshisha Women’s College of Liberal Arts, Kyoto 610-0395, Japan; dkokamoto27@yahoo.co.jp (K.O.); mkogirim@dwc.doshisha.ac.jp (M.K.); 2Department of Nutrition, Jyujyo Takeda Rehabilitation Hospital, Kyoto 601-8325, Japan; 3Department of Rehabilitation Medicine, Jyujyo Takeda Rehabilitation Hospital, Kyoto 601-8325, Japan; y-tsuji@takedahp.or.jp (Y.T.); shin.ishino@gmail.com (S.I.)

**Keywords:** convalescent rehabilitation, malnutrition, nutritional screening

## Abstract

Background/Objectives: Malnutrition significantly hinders recovery in patients undergoing convalescent rehabilitation. Proper nutritional management can improve rehabilitation outcomes. This study aimed to develop a novel nutritional screening method (J-Method) specifically in patients undergoing convalescent rehabilitation and compare it with the widely used Mini Nutritional Assessment Short Form (MNA-SF). Methods: We developed the J-Method for convalescent rehabilitation settings and compared its results with that of the MNA-SF. The J-Method comprised six items derived from various nutritional screening methods and obtained solely from medical records, without patient interviews. Data from 148 patients aged > 65 years with cerebrovascular diseases admitted to a convalescent rehabilitation ward (CRW) were collected. Nutritional status was evaluated using the J-Method and MNA-SF, after which the results were compared. Results: It is possible that the J-Method more precisely identified patients as malnourished than did the MNA-SF (J-Method: MNA-SF = 36/148 (24.3%): 111/148 (75.0%)). In detail, 75 (50.4%) were classified as having malnutrition by the MNA-SF but as non-malnutrition by the J-Method; however, no patients were in the opposite scenario. In addition, the results of nutritional screening using the J-Method identified patients in need of nutritional management intervention and suggested that to improve the rehabilitation effect, nutritional management should be initiated in an acute hospital before admission to a CRW. Conclusions: The J-Method may be more effective than the MNA-SF for nutritional screening in convalescent rehabilitation settings, as it provides a more accurate assessment of malnutrition without requiring patient interviews.

## 1. Introduction

According to the European Society for Clinical Nutrition and Metabolism (ESPEN), malnutrition is defined as “a state resulting from lack of intake or uptake of nutrition that leads to altered body composition (decreased fat-free mass) and body cell mass, causing diminished physical and mental function and impaired clinical outcomes” [1]. The goal of convalescent rehabilitation is to improve activities of daily living (ADL) and facilitate home discharge of patients with acute diseases, such as cerebrovascular and orthopedic diseases, including hip and vertebral fractures; malnutrition poses a significant barrier to this process. Over 40% of patients in convalescent rehabilitation settings experience malnutrition [2], wherein appropriate nutritional management can improve rehabilitation outcomes [3,4,5].

After admission to a convalescent rehabilitation ward (CRW), patients with malnutrition must be identified as soon as possible to ensure effective convalescent rehabilitation and nutritional management. Therefore, the first nutritional screening should be conducted within 1 or 2 days after admission [6]. Nutritional assessment is routinely performed using nutritional screening methods that can be quickly and easily performed and for which adequacy and credibility have been certified [6]. Several screening methods have been developed and are available for clinical use, each with advantages and disadvantages. The ESPEN recommends Nutritional Risk Screening 2002 (NRS-2002), the Malnutrition Universal Screening Tool, and the Mini Nutritional Assessment Short Form (MNA-SF) for evaluating hospitalized patients, community-dwelling individuals, and community-dwelling older adults, respectively [7]. The MNA-SF is considered an effective nutritional screening tool in older adults, as low MNA-SF scores can predict their functional decline [8,9,10,11,12]. However, the MNA-SF requires a medical interview, making it unsuitable for patients with impaired consciousness, cerebrovascular disease, or dementia. Another widely used tool, the Geriatric Nutritional Risk Index, assesses malnutrition risk based on body mass index (BMI) and serum albumin levels. However, it does not fully reflect the patient’s nutritional status [13], raising concerns about its suitability for patients undergoing convalescent rehabilitation. To our best knowledge, no nutritional screening method has been specifically designed for evaluating patients in CRWs.

Therefore, we propose to develop a new nutritional screening method for CRW. Because precise nutritional screening needs to be performed using several strategies [14,15], we attempted to develop a new method comprising six items extracted from several known methods (body weight [BW] loss, food intake, BMI, serum albumin concentration, decubitus, and digestive symptoms). Of the six items, BW loss, food intake, and BMI are directly related to malnutrition. Depending on the conditions, digestive symptoms, such as diarrhea, might cause malnutrition. Furthermore, treatment of decubitus should positively improve nutritional status in this patient population. Although serum albumin concentration is controversial as an index of malnutrition, it can assess general condition, including nutrition, which reflects whether patients can undergo vigorous rehabilitation to improve ADL or not. At any rate, this method uses all information from our medical records without the need for patient interviews, making it more accessible than other methods. This study presents the results of applying this method to patients in our CRW and compares its effectiveness with that of the MNA-SF. Additionally, this study discusses the potential advantages of our novel nutritional screening method (J-Method) over the MNA-SF.

## 2. Materials and Methods

### 2.1. Patients

We collected data from patients with cerebrovascular diseases aged > 65 years admitted to our CRW between February 2020 and February 2022. The following were the exclusion criteria: (1) patients who refused to participate in this study and (2) those transferred to an acute hospital due to a sudden deterioration in their health that was too severe to continue rehabilitation.

### 2.2. Establishment of a New Nutritional Screening Method (J-Method)

Based on several nutritional screening methods, we attempted to establish a new nutritional screening method (J-Method) suitable for our CRW. The J-Method comprised six items (Table 1): (1) BW loss of > 1% in a week; (2) < 50% daily food intake for 3 days after admission; (3) BMI < 18.5 kg/m^2^; (4) serum albumin concentration ≤ 3.0 g/dL; (5) the presence of decubitus anywhere in the body; and (6) miscellaneous factors, including digestive symptoms and total parenteral nutrition. A patient was diagnosed with malnutrition if they had decubitus or met more than two of the remaining criteria. The rate of BW loss was calculated using the following formula:Rate of BW loss in a week = {(BW at admission to our CRW-BW at the latest measurement in ex-hospital)/BW at the latest measurement in ex-hospital} × (7/number of days from the last measurement of BW in ex-hospital to admission to our CRW) × 100

Intake rates of principal food and side dishes in all patients were visually determined between 0 and 10 by ward nursing staff with each meal, which was recorded in the clinical records. Thus, each patient’s average daily food intake could be calculated based on meal contents (principal food and side dishes) and intake amounts for 3 days after admission by a registered dietitian. On the other hand, the height and weight of the patient were routinely measured by ward nursing staff. BMI was calculated by applying a general formula using each patient’s height (m) and weight (kg) at admission to our CRW. Serum albumin concentration was measured using blood collected from each patient during examination upon admission. Decubitus was detected through physical examination by an attending physician. Additionally, digestive symptoms, such as nausea, vomiting, and/or diarrhea, as well as total parenteral nutrition, were collected from each patient’s physician and nursing reports. Based on the J-Method’s results, the participants were assigned to the “malnutrition” and “non-malnutrition” groups.

### 2.3. Nutritional Screening Using the MNA-SF

Within 1 week of admission to our CRW, the MNA-SF was administered by KO (the first author). The MNA-SF comprised the following six items, each scored between 0 and 2 or 3: (1) food intake decline for the last 3 months [severe, 0; moderate, 1; no decrease, 3]; (2) BW loss over the last 3 months [>3 kg, 0; unknown, 1; 1–3 kg, 2; no change or increase, 3]; (3) mobility [bedridden or using wheelchair, 0; walking indoors, 1; walking outdoors, 2], (4) physical and/or mental stress [under stress, 0; no stress, 2]; (5) neurological and/or psychological diseases [severe dementia or depression, 0; moderate dementia, 1; none, 2]; and (6) BMI (<19 kg/m^2^, 0; 19–21 kg/m^2^, 1; 21–23 kg/m^2^, 2; >23 kg/m^2^, 3). Nutritional status was classified based on the total score: 0–7, 8–11, and ≥12 indicated malnutrition, risk of malnutrition, and good nutrition, respectively. Patients with an MNA-SF score < 7 were included in the malnutrition group, while the others were assigned to the non-malnutrition group.

### 2.4. Coincidence Analysis Between the J-Method and MNA-SF

Based on the J-Method and MNA-SF results, patients were divided into the following four groups. (1) The “definite malnutrition group” (D), which included patients identified as being malnourished by both the J-Method and MNA-SF. (2) The “possible malnutrition group by J-Method” (P-J), which included patients identified as being malnourished only by the J-Method. (3) The “possible malnutrition group by the MNA-SF” (P-M), which included patients identified as being malnourished only by the MNA-SF. (4) The “no malnutrition group” (N), which included patients determined as not being malnourished by both screening methods. Four items were compared across these groups: (1) food intake decline over the last 3 months, (2) intake rate of principal food and side dish in the 3 days post-admission, (3) BMI at admission to our CRW, and (4) serum albumin concentration at admission to our CRW. In addition, to verify the coincidence of the results between the J-Method and MNA-SF, the kappa coefficient (κ) was calculated. The coincidence of the results evaluated based on κ was categorized into slight, 0.0–0.20; fair, 0.21–0.40; moderate, 0.41–0.60; substantial, 0.61–0.80; and almost perfect, 0.81–1.00.

### 2.5. Sample Size Calculation and Statistical Analysis

All analyses were performed using IBM^®^SPSS Statistics 28 (IBM Corporation, Armonk, NY, USA). Normality of distribution was assessed using the Kolmogorov–Smirnov test. Regarding normally distributed data, the results are expressed as mean ± standard deviation (SD); differences among the four groups were determined using a two-way analysis of variance, with the significance of individual differences evaluated using the Tukey–Kramer post hoc test. Considering non-normally distributed data, the results are reported as medians with interquartile ranges (IQRs); differences among the four groups were analyzed using the Kruskal–Wallis test. The significance of individual differences was evaluated using the Bonferroni method as a post hoc test. *p*-values < 0.05 were considered statistically significant.

### 2.6. Ethics

This study was conducted in accordance with the Declaration of Helsinki and approved by the Ethics Committees of Doshisha Women’s College on 14 May 2020 (2020-01) and Jyujyo Takeda Rehabilitation Hospital on 18 November 2019 (20191118-1). All patients and/or their family members were informed of the study’s protocols and had the right to withdraw from the study at any time without facing any negative consequences. Personal data were fully anonymized and securely managed throughout the study.

## 3. Results

Of the 1161 patients hospitalized in our CRW between February 2020 and February 2022, there were 388 patients with cerebrovascular diseases. Of these 388, 318 patients were aged > 65 years. Among the 318, 168 patients with cerebrovascular diseases aged > 65 years agreed to participate in this study (52.8%), whereas 150 (47.2%) put off participation. However, 20 patients were excluded (5 were transferred to an acute hospital due to deterioration in physical condition and 15 lacked necessary data during hospitalization). Finally, 148 patients were included in this study (46.5% of the patients with cerebrovascular diseases aged > 65 years) (78 male and 70 female participants). This was because there were many patients who had communication problems due to dementia and aphasia and/or had no relatives. Only seven patients refused to take part in this study.

Although the patients were divided into groups (D, P-J, P-M, and N groups), there were no patients who belonged to the P-J group. Therefore, the patients were classified into only three of the four groups (D, P-M, and N). To compare BMI, its normality of distribution was verified between the three groups, and the null hypothesis was rejected with a sample size of at least 138 participants in this study with an effect size of 0.27, a detection power of 0.8, and an alpha error of 0.05.

### 3.1. Nutritional Screening Results Using the J-Method and MNA-SF

Based on nutritional screening using the J-Method, 36 and 112 patients were categorized into the malnutrition and non-malnutrition groups, respectively. Using the MNA-SF, 111 and 37 patients were categorized into the malnutrition and non-malnutrition groups, respectively. Of the patients, 37 (25.0%) and 36 (24.3%) were categorized into the non-malnutrition (N) and definite malnutrition (D) groups, respectively. Notably, 75 (50.4%) were classified as having malnutrition by the MNA-SF (P-M) but as non-malnutrition by the J-Method; however, no patients were in the opposite scenario (P-J) (Figure 1). The kappa coefficient (κ) for the coincidence between the two methods was 0.19 (Table 2), indicating that the result of nutritional screening of the J-Method slightly coincided with those of the MNA-SF.

### 3.2. Decrease in Food Intake in the Three Groups, D, P-M, and N

In the N group, 29 of 37 patients (78.4%) reported no changes in their food intake in the last 3 months before admission. However, the rate of patients who reported no changes in their food intake decreased in the P-M (44/75, 58.7%) and D (16/36, 44.4%) groups. Conversely, severe decreases in food intake in the N group were observed only in a small percentage of the individuals (2/37, 5.4%), whereas the rate of participants who reported severe decreases in food intake escalated in the D group (14/36, 38.9%), with the groups showing statistically significant differences (*p* = 0.010, Figure 2). The proportion of patients with moderate decreases in food intake was similar across the three groups.

### 3.3. Intake Rate of Principal Food and Side Dish in the Three Groups

After admission to our CRW, the patients in the N group consumed almost all of the principal food and side dishes provided by our hospital. The patients in the P-M group consumed 90%, whereas those in the D group consumed only 50% of the principal food and 65% of the side dishes, representing a significant decrease (*p* < 0.001, Table 3).

### 3.4. BMI at Admission to Our CRW in the Three Groups

The average BMI of patients at admission was 23.8 ± 2.7 kg/m^2^ in the N group, which is above the ideal BW. In the P-M group, the average BMI was 21.5 ± 3.2 kg/m^2^, which is slightly below the ideal BW but within the normal range. In the D group, the average BMI was significantly lower (18.7 ± 2.7 kg/m^2^, *p* < 0.001, Table 3), indicating low BW.

### 3.5. Concentration of Serum Albumin at Admission to Our CRW in the Three Groups

In the N and P-M groups, the average serum albumin concentrations at admission were 3.7 ± 0.4 g/dL and 3.5 ± 0.5 g/dL, respectively, showing no significant differences. However, in the D group, the average serum albumin concentration was 3.1 ± 0.7 g/dL, which is significantly lower than in the other two groups (*p* < 0.001, Table 3).

## 4. Discussion

Using the J-Method, we were able to conveniently obtain nutritional information without patient interviews, which was a key advantage. To design the J-Method, we referred to several nutritional screening methods to obtain the best results. In contrast, the MNA-SF is a widely used screening method [21] with proven validity in older adults [22,23,24]. The Global Leadership Initiative on Malnutrition (GLIM) [25] also recommends the MNA-SF as a primary nutritional screening method. Moreover, patients with malnutrition diagnosed using the MNA-SF are more likely to develop sarcopenia [26], with their severity of malnutrition being inversely related to their Barthel Index scores [27], which can show independence in ADL. Therefore, we compared the results of our J-Method with those of the MNA-SF in this study. The kappa coefficient (κ = 0.19) indicated slight agreement between the two methods. Several factors may explain this. All patients classified as having malnutrition by the J-Method were also categorized as such by the MNA-SF (the D group). However, no patients were identified as having malnutrition by the J-Method but not by the MNA-SF. Moreover, >50% of patients were classified as having malnutrition by the MNA-SF only. This may be due to MNA-SF’s inclusion of items like “walking independently” and “experience of mental stress and/or acute diseases during the last 3 months”. Most patients are admitted to our CRW after acute diseases, such as cerebrovascular diseases and/or orthopedic disorders, including femoral neck fracture and vertebral compression fracture. In other words, most of the patients undergoing convalescent rehabilitation have experienced acute diseases and cannot not walk independently at the time of admission to our CRW. Thus, when the MNA-SF is used in these patients, it is likely that the MNA-SF scores are low and that the patients are determined as having malnutrition. The MNA-SF has recently been reported to have high sensitivity but low specificity [28], making it a less suitable nutritional screening method for evaluating patients with convalescent rehabilitation. In contrast, 24.3% of patients were diagnosed as having malnutrition by the J-Method. As mentioned above, >40% of patients in rehabilitation settings may suffer from malnutrition. Although our results were lower than expected, the J-Method showed potential as a specialized nutritional screening method for convalescent rehabilitation patients.

Although food intake over the last 3 months before admission did not change in >50% of the patients in the P-M group, it moderately or severely decreased in >50% of the patients in the D group. These trends persisted after admission, as the D group consumed significantly less of the principal food and side dishes provided by the hospital than the N and P-M groups. This result suggests that the patients’ food intake declined gradually before admission rather than immediately upon admission. Moreover, the patients in the D group had lower BMIs and serum albumin concentrations than those in the N and P-M groups. It has been reported that low BMI and/or serum albumin concentration diminishes the effect of convalescent rehabilitation. Therefore, identifying patients who have malnutrition accurately and intervening early is crucial; however, improvements in food intake should also be prioritized in the acute hospital setting.

The question items in the J-Method were extracted from several nutritional screening methods recommended by the nutritional assessment guidelines (Table 4), with each item being individually significant in those screening methods. When performing nutritional screening, combining several methods is recommended [13]. This indicates that by combining various nutritional screening methods, we could develop a novel and effective tool, such as the J-Method. To verify the efficacy of the J-Method, it was initially tested only in our rehabilitation setting. We do not consider the J-Method the best method for the CRW; its question items may be revised in the future. Regarding broader application, the J-Method should be tentatively implemented in other institutions, such as associated facilities in our group, at the first onset. On the other hand, to apply this method in all patients in CRW, the nutritional status of the patients with diseases other than cerebrovascular diseases in CRW must be assessed using the J-Method. In addition, we have to recognize that 53.5% of the patients with cerebrovascular diseases in our CRW did not participate in this study, which indicates that the results of this study have limitations. Nonetheless, the J-Method has significant potential for improving the evaluation of nutritional status in patients undergoing convalescent rehabilitation. 

## 5. Conclusions

In conclusion, we developed the J-Method specifically for convalescent rehabilitation settings and compared its results with those of the MNA-SF. In these settings, the J-Method might prove to be more effective than the MNA-SF. Continuous efforts to refine nutritional screening methods will help improve nutrition management in patients undergoing convalescent rehabilitation.

## Figures and Tables

**Figure 1 nutrients-16-03997-f001:**
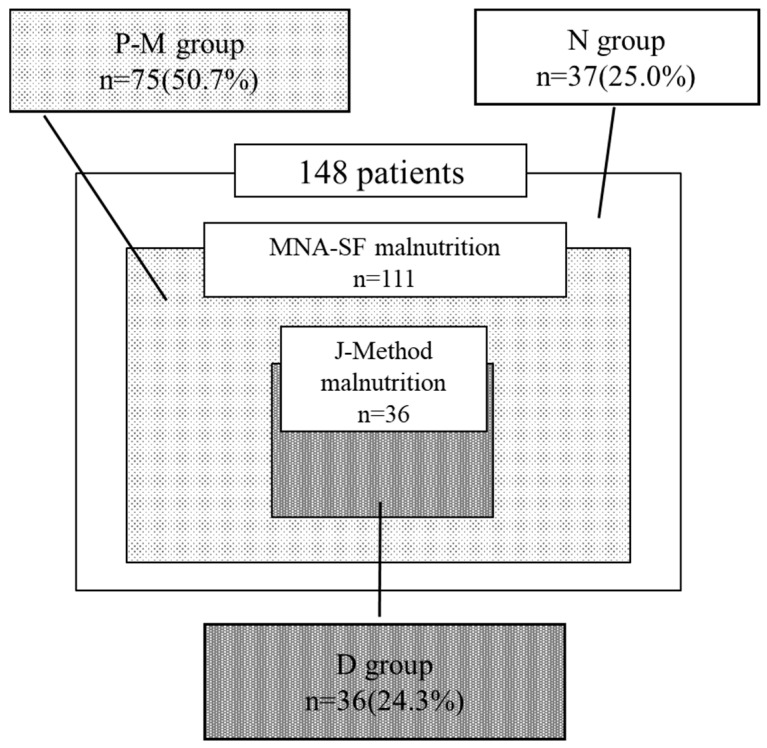
The results of nutritional screening using the J-Method and MNA-SF. The J-Method and MNA-SF identified 36 and 111 patients as having malnutrition, respectively. Although 37 patients (25.0%) are classified as “N”, 36 (24.3%) are classified as “D” when combining the J-Method and MNA-SF. Notably, 75 patients (50.4%) are classified as having malnutrition by the MNA-SF but as non-malnutrition by the J-Method (P-M).

**Figure 2 nutrients-16-03997-f002:**
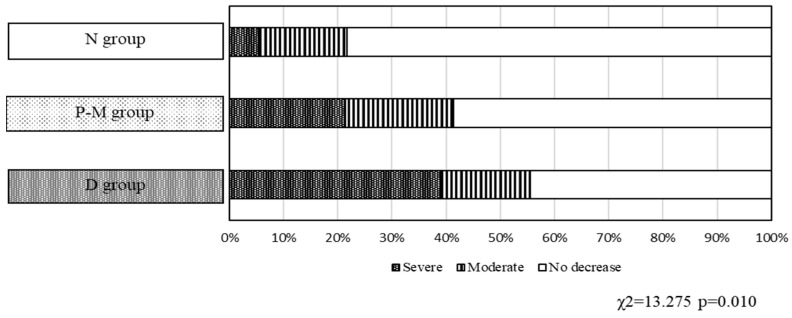
Food intake in the N, P-M, and D groups. The percentages of patients who reported no changes in food intake over the last three months before admission were 78.4% (29/37 patients), 58.7% (44/75 patients), and 44.4% (16/36 patients) in the N, P-M, and D groups, respectively. Significant differences were observed between the three groups (*p* = 0.010).

**Table 1 nutrients-16-03997-t001:** Six screening items in the J-Method and their references.

J-Method	References
Body weight loss of > 1% in a week	Body weight loss rate of 5% per month [16]
Less than 50% of daily food intake for 3 days after admission	Dietary intake rate of 50% in the NRS2002 [16]
BMI < 18.5 kg/m^2^	Diagnostic criterion of low body weight [17]
Serum albumin concentration up to 3.0 g/dL	Criterion for NST intervention [18]
Presence of decubitus in the body	Active nutritional therapy required [19]
Digestive symptoms and total parenteral nutrition	Factors leading to malnutrition [20]

BMI, body mass index; NST, nutrition support team; NRS2002, Nutritional Risk Screening 2002.

**Table 2 nutrients-16-03997-t002:** The results of nutritional screening using the J-Method and MNA-SF.

		MNA-SF		Total
		Malnutrition	Non-Malnutrition	
J-Method	malnutrition	36	0	36
	non-malnutrition	75	37	112
	Total	111	37	148
				κ = 0.19

J-Method, novel nutritional screening method; MNA-SF, Mini Nutritional Assessment Short Form.

**Table 3 nutrients-16-03997-t003:** Physical characteristics and food intake rates at hospitalization in the three groups.

Total (N = 148)	N Group (n = 37)	P-M Group (n = 75)	D Group (n = 36)	*p* Value
	F/M = 14/23	F/M = 33/42	F/M = 23/13	
Age (years old)	76.9 ± 7.1 ^a^	76.7 ± 6.5 ^a^	80.7 ± 6.9 ^b^	0.013
BMI (kg/m^2^)	23.8 ± 2.7 ^a^	21.5 ± 3.2 ^b^	18.7 ± 2.7 ^c^	<0.001
Serum albumin concentration (g/dL)	3.7 ± 0.4 ^a^	3.5 ± 0.5 ^a^	3.1 ± 0.7 ^b^	<0.001
Intake rate of principal food (%)	100 (90–100) ^a^	90 (80–100) ^a^	50 (33–98) ^b^	<0.001
Intake rate of side dish (%)	100 (80–100) ^a^	90 (80–100) ^a^	65 (40–90) ^b^	<0.001

Mean ± SD or median (IQR). There is a significant difference between the different letters. The average BMI, serum albumin concentration, and food intake rate in the D group were significantly lower than those in the other groups (*p* < 0.001).

**Table 4 nutrients-16-03997-t004:** Combination of several nutritional screening methods recommended by the nutrition assessment guidelines.

Screening Method	Rate of BW Loss	Decrease in FI	BMI	BW/IBW	Acute Disease	Serum Albumin	TLC	TC	Digestive Symptoms	TPN	Decubitus
GNRI				○		○					
CONUT						○	○	○			
MST	○										
MUST	○	○	○								
NRS2002	○	○	○		○						
J-Method	○	○	○	Substituted by BMI		○			○	○	○

BW, body weight; FI, food intake; BMI, body mass index; IBW, ideal body weight; TLC, total lymphocyte count; TC, total cholesterol; TPN, total parenteral nutrition; GNRI, Geriatric Nutritional Risk Index; CONUT, Controlling Nutritional Status; MST, Malnutrition Screening Tool; MUST, Malnutrition Universal Screening Tool; NRS2002, Nutritional Risk Screening 2002.

## Data Availability

The original contributions presented in the study are included in the article, further inquiries can be directed to the corresponding author.

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
