# Peer review of "Establishment of a Rapid and Precise Nutritional Screening Method for Convalescent Rehabilitation Patients: A Preliminary Study"

_nutrients, 2024, doi:10.3390/nu16233997_

Round 1
Reviewer 1 Report
Comments and Suggestions for Authors
this is a very interesting research, but to develop a new tool you have to follow similar steps to depvelos a questtionarie, in order to create it with a delphi panel, and test psycometric properties.
author should develop the standard methodology to validate a new assesment tool, an example documentation can be found here:https://www.ncbi.nlm.nih.gov/books/NBK560531/#:~:text=The%20validity%20of%20an%20assessment,integrity%20of%20the%20assessment%20approach.
Author Response
Comment: This is a very interesting research, but to develop a new tool you have to follow similar steps to develop a questionnaire, in order to create it with a delphi panel, and test psychometric properties.
Author should develop the standard methodology to validate a new assesment tool, an example documentation can be found:
here:https://www.ncbi.nlm.nih.gov/books/NBK560531/#:~:text=The%20validity%20of%20an%20assessment,integrity%20of%20the%20assessment%20approach.
Response: We thank you for this comment and your encouraging words. As you kindly pointed out, we agree with you that referring to this panel assists in developing a questionnaire. However, our novel method was developed based on extracting question items from several nutritional screening methods recommended by the nutritional assessment guidelines. In our future research we will follow the methodology you excellently provided. Once again, thank you for your suggestion.
Reviewer 2 Report
Comments and Suggestions for Authors
I have read and reviewed the paper by Kozue Okamoto et al. titled "Development of a quick and accurate nutritional screening test for rehabilitation patients post recovery stage". The manuscript provides a highly valuable finding regarding the efficacy of a newly developed J-Method in the identification of rehabilitation patients at the risk of malnutrition, particularly compared to the MNA-SF. It stresses the convenience of the J-Method, which only uses medical records, thus constituting a convenient alternative in the examination of those who cannot be interviewed. However, I would like to tell that there are few points for improvement. a. First is the clear expression of rationale that justifies why a particular criterion was selected specifically in the J-Method. This can explain why a particular parameter was favored more than others. This should be done in Introduction section itself. b. The second one is a more widened discussion on its limitations, especially its generalizability to other patient populations. c. The statistical section should clearly indicate whether power analysis was performed to see if it will provide sufficient sample size for detecting any significant difference. Taking care of these points would add strength to the manuscript by providing more reasonable context and being more transparent about the methodology and its applicability. Authors should do necessary changes and resubmit the manuscript.
Author Response
Comment: I have read and reviewed the paper by Kozue Okamoto et al. titled "Development of a quick and accurate nutritional screening test for rehabilitation patients post recovery stage". The manuscript provides a highly valuable finding regarding the efficacy of a newly developed J-Method in the identification of rehabilitation patients at the risk of malnutrition, particularly compared to the MNA-SF. It stresses the convenience of the J-Method, which only uses medical records, thus constituting a convenient alternative in the examination of those who cannot be interviewed. However, I would like to tell that there are few points for improvement.
a) First is the clear expression of rationale that justifies why a particular criterion was selected specifically in the J-Method. This can explain why a particular parameter was favored more than others. This should be done in Introduction section itself.
b) The second one is a more widened discussion on its limitations, especially its generalizability to other patient populations.
c) The statistical section should clearly indicate whether power analysis was performed to see if it will provide sufficient sample size for detecting any significant difference. Taking care of these points would add strength to the manuscript by providing more reasonable context and being more transparent about the methodology and its applicability. Authors should do necessary changes and resubmit the manuscript.
Response: We thank for your constructive comments.
a) According to your suggestion, the sentences were added in Introduction as follows:
Of the 6 items, BW loss, food intake, and BMI would be directly related to malnutrition. Depending on conditions, the digestive symptoms, such as diarrhea, might cause malnutrition. Furthermore, treatment of decubitus should positively improve nutritional status in this patient population. Although serum albumin concentration is controversial as an index of malnutrition, it could assess general condition including nutrition, which reflects whether patients could undergo vigorous rehabilitation to improve ADL or not. (Line 67-73)
b) According to your suggestion, some discussions were added as follows:
Regarding broader application, the J-Method should be tentatively implemented in other institutions, such as associated facilities in our group, at the first onset. On the other hand, to apply this method in all patients in CRW, the nutritional status in the patients with diseases except for cerebrovascular diseases in CRW must be assessed by the J-Method. In addition, we have to recognize that 53.5% of the patients with cerebrovascular diseases in our CRW did not participate in this study, which indicated that the results of this study have limitations. (Line 279-285)
c) Sample size has been determined as follows:
Although the patients were divided into groups (D, P-J, P-M, and N groups), there were no patients who belonged to the P-J group. Therefore, the patients were classified into only three of the four groups (D, P-M, and N). To compare BMI, its normality of distribution was verified, between the three groups, the null hypothesis was rejected with a sample size of at least 138 participants in this study with an effect size of 0.27, a detection power of 0.8, and an alpha error of 0.05. (Line 167-172)
Reviewer 3 Report
Comments and Suggestions for Authors
This study aimed to develop and evaluate a new nutritional screening method, the J-Method, for patients undergoing convalescent rehabilitation, comparing its effectiveness to the Mini Nutritional Assessment Short Form (MNA-SF). The J-Method, which consists of six items derived from various nutritional screening methods and relies solely on medical records, was tested on 148 patients over 65 with cerebrovascular diseases in a rehabilitation ward. Results showed that the J-Method more accurately identified malnourished patients than the MNA-SF, highlighting the need for nutritional management interventions to begin in acute hospital settings prior to rehabilitation. Overall, the J-Method may offer a superior approach to nutritional screening in convalescent rehabilitation by providing a more precise assessment without patient interviews.
It would be helpful to discuss any limitations of the J-Method, such as its reliance on medical records, which might miss certain nuances of patient health that interviews could uncover.
Author Response
Comments: This study aimed to develop and evaluate a new nutritional screening method, the J-Method, for patients undergoing convalescent rehabilitation, comparing its effectiveness to the Mini Nutritional Assessment Short Form (MNA-SF). The J-Method, which consists of six items derived from various nutritional screening methods and relies solely on medical records, was tested on 148 patients over 65 with cerebrovascular diseases in a rehabilitation ward. Results showed that the J-Method more accurately identified malnourished patients than the MNA-SF, highlighting the need for nutritional management interventions to begin in acute hospital settings prior to rehabilitation. Overall, the J-Method may offer a superior approach to nutritional screening in convalescent rehabilitation by providing a more precise assessment without patient interviews.
It would be helpful to discuss any limitations of the J-Method, such as its reliance on medical records, which might miss certain nuances of patient health that interviews could uncover.
Response: We thank for your helpful comments. But, if we discuss the reliance on medical record, J-Method would crumble at its foundation. Instead of this, we discussed the limitation of J-Method in Discussion as follows:
Regarding broader application, the J-Method should be tentatively implemented in other institutions, such as associated facilities in our group, at the first onset. On the other hand, to apply this method in all patients in CRW, the nutritional status in the patients with diseases except for cerebrovascular diseases in CRW must be assessed by the J-Method. In addition, we have to recognize that 53.5% of the patients with cerebrovascular diseases in our CRW did not participate in this study, which indicated that the results of this study have limitations. (Line 279-285)
Reviewer 4 Report
Comments and Suggestions for Authors
Overall, the paper is well organized and constructed with clear headings and subheadings that make it easy to understand the methodology and results. Each paragraph advances logically, enabling the reader to comprehend the study's design and findings. However, the evidence presented merits some comments and raises some unresolved questions.
Regarding the results of the study that are mentioned in the abstract (lines 20 to 24), more numerical data could be written to robustly support the conclusion that the method is effective in the screening of malnutrition in those patients.
Regarding the paper’s introduction some modifications could be made. Firstly, the use of the term “establish” in line 13 of the introduction could be replaced by the term “develop”, since that is what the aim of the presented study is. In lines 20 and 21 the author is referring that “The J-Method more precisely identified patients as malnourished than the MNA-SF” a result that cannot be obtained by the study’s methodology considering that to make assumptions of the precision of a tool, the study’s results should be repeated under similar circumstances. In lines 43 and 61, it might be more precise to use the term “screening” instead of “assessment” because the study is about the development of a nutritional screening tool. As mentioned above, it would be reasonable in line 60 the term “establishing” to be replaced by “developing”.
With regard to the methodology of the paper, the patient selection criteria are well stated, which is critical for comprehending the study population and assuring reproducibility. Nevertheless, more details about the sample size and inclusion and exclusion criteria may be needed. Additional information about the Body Mass Index (BMI) and its formula should me mentioned in the methodology section. To enhance credibility of the calculations of BMI and rate of body weight loss references must be added. The description of the J-Method for nutritional screening is extensive. More information about the methodology of the extraction of the 6 items the J-method consists of is needed. Furthermore, it would be prudent for a pilot test to be performed in order to investigate the feasibility and efficacy of J-method. The main challenge about malnutrition uncovering is that in a plethora of cases, nutritional screening and assessment tools are not used by health care professionals because they are not easy to use and, in most cases, time consuming. Such tools should be easy to use, quick, economical, standardized, and validated. Furthermore, a more detailed report should be made about by whom and how data (weight, height, average daily food intake) were collected. Questions about the proper collection of the data may arise since inappropriate data collection could lead to biases. It is of vital importance to note that neither the reliability nor validity of the tool were tested.
The normality test conducted to distinguish between normally and non-normally distributed data should have been noted in the statistical analysis.
With respect to the presentation of the results, a descriptive statistics table would be wise to have been constructed and added in the paper. This way, it would be by far easier for the readers to learn about the main features of the 3 groups of our study (D group, N group and P-M group) and if there were any statistically significant differences between the groups (lines 145-193). Also, I would suggest lines 166-169 to be paraphrased for better comprehension. For example, “Severe decreases in food intake in the N group were observed only in a small percentage of the subjects (2/37,5%), while the rate of participants who reported severe decreases in food intake escalated in the D group (14/36, 38,9%), with the groups presenting a statistically significant difference (p = 0.010, Figure 2). The proportion of patients with moderate decreases in food intake was similar across the three groups.”
In the Discussion and specifically in line 206 the term “establish” it would be more precise to be replaced by the term “design”.
To summarize, the paper presents a valuable investigation into nutritional screening methods in older patients with cerebrovascular diseases in rehabilitation facilities. The findings could contribute significantly to improving patient care in rehabilitation settings. Τhe limitations of the study and potential biases as well as suggestions for future research that could build upon your findings, must be mentioned.
Author Response
Comments 1: Overall, the paper is well organized and constructed with clear headings and subheadings that make it easy to understand the methodology and results. Each paragraph advances logically, enabling the reader to comprehend the study's design and findings. However, the evidence presented merits some comments and raises some unresolved questions.
Regarding the results of the study that are mentioned in the abstract (lines 20 to 24), more numerical data could be written to robustly support the conclusion that the method is effective in the screening of malnutrition in those patients.
Response 1: According to your suggestion, Abstract was modified as follows:
Results: It is possible that the J-Method more precisely identified patients as malnourished than did the MNA-SF (J-Method: MNA-SF = 36/148 (24.3%): 111/148 (75.0%)). In details, 75 (50.4%) were classified as having malnutrition by the MNA-SF but as non-malnutrition by the J-Method; however, no patients were in the opposite scenario. (Line 20-24)
Comments 2: Regarding the paper’s introduction some modifications could be made. Firstly, the use of the term “establish” in line 13 of the introduction could be replaced by the term “develop”, since that is what the aim of the presented study is. In lines 20 and 21 the author is referring that “The J-Method more precisely identified patients as malnourished than the MNA-SF” a result that cannot be obtained by the study’s methodology considering that to make assumptions of the precision of a tool, the study’s results should be repeated under similar circumstances. In lines 43 and 61, it might be more precise to use the term “screening” instead of “assessment” because the study is about the development of a nutritional screening tool. As mentioned above, it would be reasonable in line 60 the term “establishing” to be replaced by “developing”.
Response 2: According to your suggestion, terms were changed. And, we changed the sentence in Line 20-21 as follows:
It is possible that the J-Method more precisely identified patients as malnourished than did the MNA-SF (J-Method: MNA-SF = 36/148 (24.3%): 111/148 (75.0%)). (Line 20-21)
Comments 3: With regard to the methodology of the paper, the patient selection criteria are well stated, which is critical for comprehending the study population and assuring reproducibility. Nevertheless, more details about the sample size and inclusion and exclusion criteria may be needed. Additional information about the Body Mass Index (BMI) and its formula should me mentioned in the methodology section. To enhance credibility of the calculations of BMI and rate of body weight loss references must be added. The description of the J-Method for nutritional screening is extensive. More information about the methodology of the extraction of the 6 items the J-method consists of is needed. Furthermore, it would be prudent for a pilot test to be performed in order to investigate the feasibility and efficacy of J-method. The main challenge about malnutrition uncovering is that in a plethora of cases, nutritional screening and assessment tools are not used by health care professionals because they are not easy to use and, in most cases, time consuming. Such tools should be easy to use, quick, economical, standardized, and validated. Furthermore, a more detailed report should be made about by whom and how data (weight, height, average daily food intake) were collected. Questions about the proper collection of the data may arise since inappropriate data collection could lead to biases. It is of vital importance to note that neither the reliability nor validity of the tool were tested.
Response 3: We thank for your advice.
Sample size has been determined as follows:
Although the patients were divided into groups (D, P-J, P-M, and N groups), there were no patients who belonged to the P-J group. Therefore, the patients were classified into only three of the four groups (D, P-M, and N). To compare BMI, its normality of distribution was verified, between the three groups, the null hypothesis was rejected with a sample size of at least 138 participants in this study with an effect size of 0.27, a detection power of 0.8, and an alpha error of 0.05. (Line 167-172)
Regarding six questions in J-Method, the reason why the questions were selected was added with Table 1 as follows:
Of the 6 items, BW loss, food intake, and BMI would be directly related to malnutrition. Depending on conditions, the digestive symptoms, such as diarrhea, might cause malnutrition. Furthermore, treatment of decubitus should positively improve nutritional status in this patient population. Although serum albumin concentration is controversial as an index of malnutrition, it could assess general condition including nutrition, which reflects whether patients could undergo vigorous rehabilitation to improve ADL or not. (Line 67-73)
Regarding the pilot test of J-Method, we proposed as follows:
Regarding broader application, the J-Method should be tentatively implemented in other institutions, such as associated facilities in our group, at the first onset. (Line 279-281)
Regarding the way to collect the data of BW, height and food intake, the method was added in Materials and Method as follows:
Intake rates of principal food and side dishes in all patients were visually determined between 0 and 10 by ward nursing staff with each meal, which was recorded in the clinical records. Thus, each patient's average daily food intake could be calculated based on meal contents (principal food and side dishes) and intake amounts for 3 days after admission by registered dietitian. On the other hand, height and weight of patient were routinely measured by ward nursing staff: BMI was calculated by applying a general formula using each patient’s height (m) and weight (kg) at admission to our CRW. (Line 95-101)
Comments 4: The normality test conducted to distinguish between normally and non-normally distributed data should have been noted in the statistical analysis.
Response 4: Regarding the normality test, the sentence was added in Materials and Method as follows:
Normality of distribution was assessed using Kolmogorov-Smirnov test. (Line 140)
Comments 5: With respect to the presentation of the results, a descriptive statistics table would be wise to have been constructed and added in the paper. This way, it would be by far easier for the readers to learn about the main features of the 3 groups of our study (D group, N group and P-M group) and if there were any statistically significant differences between the groups (lines 145-193). Also, I would suggest lines 166-169 to be paraphrased for better comprehension. For example, “Severe decreases in food intake in the N group were observed only in a small percentage of the subjects (2/37,5%), while the rate of participants who reported severe decreases in food intake escalated in the D group (14/36, 38,9%), with the groups presenting a statistically significant difference (p = 0.010, Figure 2). The proportion of patients with moderate decreases in food intake was similar across the three groups.”
Response 5: According to your suggestion, we newly made Table 3, in which the results were described. Moreover, the sentence was changed as follows:
Severe decreases in food intake in the N group were observed only in a small percentage of the individuals (2/37, 5.4%), whereas the rate of participants who reported severe decreases in food intake escalated in the D group (14/36, 38.9%), with the groups showing statistically significant differences (p = 0.010, Figure 2). (Line 194-197)
Comments 6: In the Discussion and specifically in line 206 the term “establish” it would be more precise to be replaced by the term “design”.
Response 6: According to your suggestion, the term was changed.
Comments 7: To summarize, the paper presents a valuable investigation into nutritional screening methods in older patients with cerebrovascular diseases in rehabilitation facilities. The findings could contribute significantly to improving patient care in rehabilitation settings. Τhe limitations of the study and potential biases as well as suggestions for future research that could build upon your findings, must be mentioned.
Response 7: According to your suggestion, the Discussion was rewritten as follows:
To verify the efficacy of the J-Method, it was initially tested only in our rehabilitation setting. We do not consider the J-Method the best method for the CRW; its question items may be revised in the future. Regarding broader application, the J-Method should be tentatively implemented in other institutions, such as associated facilities in our group, at the first onset. On the other hand, to apply this method in all patients in CRW, the nutritional status in the patients with diseases except for cerebrovascular diseases in CRW must be assessed by the J-Method. In addition, we have to recognize that 53.5% of the patients with cerebrovascular diseases in our CRW did not participate in this study, which indicated that the results of this study have limitations. (Line 277-285)
Round 2
Reviewer 1 Report
Comments and Suggestions for Authors
your explnataions are right, in this case explain in title that this is an preliminary study, due in next studies you should develop the tool validation
Author Response
Comments: your explnataions are right, in this case explain in title that this is an preliminary study, due in next studies you should develop the tool validation
Response: According to your suggestion, we changed the title of our manuscript as follows:
Establishment of a rapid and precise nutritional screening method for convalescent rehabilitation patients-Preliminary Study-
Reviewer 2 Report
Comments and Suggestions for Authors
The authors have answered all queries. The article may be accepted in current form.
Author Response
Comments: The authors have answered all queries. The article may be accepted in current form.
Response: Thank you for your comment. We appreciate your decision.